# Optimal learning rate scaling depends on data in deep scalar linear networks

**Yedi Zhang[1]    Peter E. Latham[1]    Leena Chennuru Vankadara[1]    Andrew Saxe[1,2]**

[1]Gatsby Computational Neuroscience Unit, University College London
[2]Sainsbury Wellcome Centre, University College London
`{yedi,pel}@gatsby.ucl.ac.uk,{l.vankadara,a.saxe}@ucl.ac.uk`

## Abstract

We study the gradient descent dynamics of deep scalar linear networks, $f(x) = \prod_{l=1}^{L} w_l x$, which enjoy exact time-course solutions for any integer depth. We show that even in this minimal model, the optimal depth-wise learning rate scaling depends on data, whereas data-agnostic scaling rules fail to transfer across depths. Under the data-dependent optimal scaling, the learning dynamics is independent of data and weakly dependent on depth, resulting in a constant linear convergence rate across all depths including infinity. We further show similar data-dependent effects in deep scalar linear networks with residual connections.

## 1 Introduction

The large scale of modern neural networks has been empirically shown to play a crucial role in the rapid progress of deep learning models (Kaplan et al., 2020; Hoffmann et al., 2022). One essential factor of the scale is depth. Thus, understanding how to enable hyperparameter transfer across depth is critical for achieving predictable gains from scale. While existing literature on hyperparameter transfer suggests that data-agnostic learning rate scaling can allow depth-wise transfer (Yang et al., 2024; Everett et al., 2024; Noci et al., 2024; Bordelon et al., 2024b;a; Bordelon & Pehlevan, 2025), we demonstrate that even in a minimal model class of deep scalar linear networks, the optimal learning rate scaling is inherently data-dependent. We show that under the data-dependent scaling, the learning rate transfers across depth, whereas data-agnostic scaling does not enable transfer.

We consider the simplest possible deep network, a depth-$L$ scalar linear chain defined as

$$f(x; w) = \prod_{l=1}^{L} w_l x, \quad x, w_1, \cdots, w_L \in \mathbb{R}. \tag{1}$$

Building on and refining analyses in prior work (Saxe et al., 2014; 2019), we write exact solutions to the full gradient descent learning dynamics for any integer depth, expressed via special functions, i.e. the hypergeometric function and the Lambert $W$ function. Under the data-dependent optimal learning rate scaling and a balanced initialization scheme, the gradient descent learning dynamics is independent of data and weakly dependent on depth. This results in a constant linear convergence rate across all depths, including the limiting case of infinite depth. Further, we extend the analysis to deep scalar linear residual networks with block depth one and two, and find that the optimal learning rate scaling for them is also data-dependent.

**Related work**. Jelassi et al. (2023) found that in deep ReLU networks with mean-field initialization, the largest learning rate for which the changes in the pre-activations after one gradient descent step remains bounded scales with depth $L$ as $L^{-3/2}$. Bordelon et al. (2024b); Bordelon & Pehlevan (2025) obtained reduced learning dynamics and studied hyperparameters transfer in infinite-depth linear residual networks in early training time, where the width and depth limits commute (Hayou & Yang, 2023). Dey et al. (2025) demonstrated that deep residual networks with $L^{-1/2}$ scaling can achieve hyperparameter transfer but operate in a locally lazy learning regime, while a $L^{-1}$ scaling enables rich learning and depth-wise hyperparameter transfer. Complementing these findings, we use a simple model class of deep scalar linear networks to demonstrate that the optimal learning rate scaling is data-dependent.

The learning dynamics of deep linear networks enjoy a rich line of theoretical results (Baldi & Hornik, 1989; Fukumizu, 1998; Saxe et al., 2014; 2019; Arora et al., 2018; Shamir, 2019; Lampinen & Ganguli, 2019; Gidel et al., 2019; Advani et al., 2020; Huh, 2020; Gissin et al., 2020; Tarmoun et al., 2021; Atanasov et al., 2022; Braun et al., 2022; Shi et al., 2022; Zhang et al., 2024; Dominé et al., 2025; Xu & Ziyin, 2025; Watanabe et al., 2026). Saxe et al. (2014; 2019) solved the learning dynamics of deep linear networks with aligned small initial weights and white input covariance, showing that depth slows down learning in the case of learning with $\ell_2$ loss and the infinite-depth network incurs a finite decay in learning speed relative to the shallow network. Here we build on and refine these results by incorporating input correlations, expressing solutions via special functions, and choosing variables that reveal data-independent learning dynamics, and connect these results to the modern literature on hyperparameter transfer.

## 2  LEARNING DYNAMICS WITH OPTIMAL LEARNING RATE SCALING

Let $\{x_n, y_n\}_{n=1}^N$ be a training set. The gradient flow dynamics of the depth-$L$ linear chain in Equation (1) trained with $\ell_2$ loss, $\mathcal{L} = \frac{1}{2N} \sum_{n=1}^N (y - f(x))^2$, is given by

$$\dot{w}_l = -\eta \frac{\partial \mathcal{L}}{\partial w_l} = \eta \left( \mu_{yx} - \mu_{xx} \prod_{i=1}^L w_i \right) \prod_{i \neq l} w_i, \tag{2}$$

where $\mu_{yx} = \frac{1}{N} \sum_{n=1}^N y_n x_n, \mu_{xx} = \frac{1}{N} \sum_{n=1}^N x_n^2$ are moments of the dataset, and $\eta$ represents the learning rate. The continuous-time gradient flow dynamics captures the behaviors of gradient descent under a stable learning rate where the dynamics does not diverge or sustainedly oscillate (Cohen et al., 2025). We consider the stable regime of gradient descent learning in this paper. Here we present a self-contained exposition that builds on Saxe et al. (2014; 2019), incorporating several refinements.

The dynamics in Equation (2) admits a well-known conservation law (Fukumizu, 1998; Saxe et al., 2014; Du et al., 2018) between any pairs of weights, $\frac{d}{dt} \left( w_l^2 - w_{l'}^2 \right) = 0$. If we assume all initial weights are positive[1], $w_l(0) > 0 \, \forall l$, we can use the conservation law to reduce the $L$-dimensional dynamics in Equation (2) to an one-dimensional ordinary differential equation about $w_1$

$$\dot{w}_1 = \eta \left( \mu_{yx} - \mu_{xx} \prod_{i=1}^L \sqrt{w_1^2 + c_i} \right) \prod_{i=2}^L \sqrt{w_1^2 + c_i}, \quad \text{where } c_l = w_l(0)^2 - w_1(0)^2. \tag{3}$$

We further assume that the initial weights of all layers are equal, $w_l(0) = w_1(0) \, \forall l$. This is motivated by the fact that we typically want all layers to participate in learning in a balanced way. Due to the conservation law, the weights that are initialized equal will remain equal throughout training. With the equal initial weight assumption, the dynamics in Equation (3) simplifies to

$$\dot{w}_1 = \eta \left( \mu_{yx} - \mu_{xx} w_1^L \right) w_1^{L-1}. \tag{4}$$

**Maximum stable learning rate**. When we increase the depth of the linear network, the global minimum of the loss landscape becomes sharper, as shown in Figure 1A. The second-order derivative, i.e. the sharpness, at the global minimum is $S = \mu_{xx} L \left( \frac{\mu_{yx}}{\mu_{xx}} \right)^{2-2/L}$. A sharper minimum requires a smaller learning rate for gradient descent dynamics to be stable. In particular, the learning rate should satisfy: $0 < \eta < 2/S$. Hence, the maximum stable learning rate scales as

$$\eta = \tau^{-1} \frac{1}{\mu_{xx} L} \left( \frac{\mu_{yx}}{\mu_{xx}} \right)^{-2+2/L}, \tag{5}$$

where $\tau \in (0.5, \infty)$ is the time constant. The gradient descent dynamics exhibits oscillations when $\tau \leq 0.5$, as shown in Figure 1C. Equation (5) shows that even in this minimal setup, the scaling of the maximum stable learning rate is data-dependent, with implications on hyperparameter transfer that we will examine in Section 3.

**Dynamics with maximum stable learning rate**. We now analyze the gradient flow dynamics with the maximum stable learning rate. We are interested in how the total weight evolves to approach the

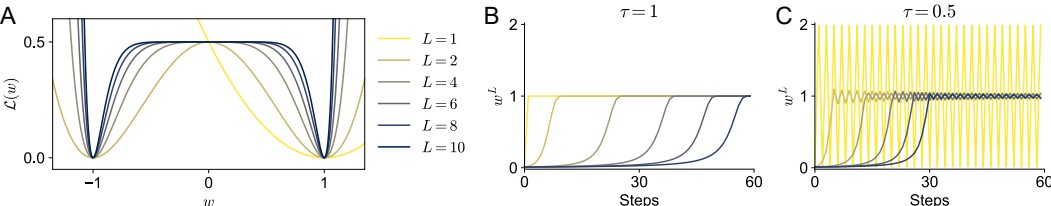

Figure 1: The loss landscape of a scalar linear network has a sharper global minimum as the depth $L$ increases, requiring a smaller learning rate for stable gradient descent dynamics. (A) Plot of the loss function $\mathcal{L}(w) = \left(1 - w^L\right)^2/2$ with different $L$. (B) Gradient descent trajectory of the total weight $w^L$ using learning rates that scale as Equation (5) with $\tau = 1$. The dynamics is stable. (C) Same as panel B but with $\tau = 0.5$, which is the threshold for stable gradient descent dynamics. The dynamics exhibits oscillations.

target weight over training. We thus study the dynamics of the ratio between the total weight and the target weight, $\alpha(t) = w_1(t)^L \mu_{xx}/\mu_{yx}$, which evolves as

$$\tau\dot{\alpha} = \alpha^{2-2/L}\left(1 - \alpha\right). \tag{6}$$

By using the maximum stable learning rate scaling and tracking the relative total weight rather than weights of individual layers, we obtain an ordinary differential equation (6) that is independent of the data statistics and weakly dependent on the depth $L$, i.e. through the factor $\alpha^{2-2/L}$. The dependence on $L$ weakens as $L$ increases, since $\lim_{L \to \infty} \alpha^{2-2/L} = \alpha^2$.

**Exact time-course solution**. Equation (6) is a separable differential equation. By separating variables and integrating both sides, we obtain the solution of $t$ in terms of $\alpha$ for any positive integer depth $L$

$$t = \tau \frac{\alpha^{\frac{2}{L}-1}}{\frac{2}{L}-1} {}_2F_1\left(1, \frac{2}{L}-1; \frac{2}{L}; \alpha\right)\Bigg|_{\alpha(0)}^{\alpha(t)}, \tag{7}$$

where ${}_2F_1$ is the hypergeometric function. For a general integer $L$, we cannot invert Equation (7) to solve $\alpha$ in terms of time $t$ due to the intractability of the hypergeometric function as a special function. However, we can invert Equation (7) for several specific depths, $L = 1, 2, \infty$, in which the hypergeometric function reduces to elementary functions. Specifically, in the limit of infinite-depth $L \to \infty$, the dynamics of $\alpha$ is given by

$$\tau\dot{\alpha} = \alpha^2\left(1 - \alpha\right). \tag{8}$$

The solution to Equation (8) can be expressed as

$$\alpha(t) = \frac{1}{1 + W_0(e^{\beta(t)})}, \quad \text{where } \beta(t) = -\frac{t}{\tau} + \frac{1}{\alpha_0} + \ln\left(\frac{1}{\alpha_0} - 1\right) - 1, \quad 0 < \alpha \leq 1. \tag{9}$$

Here $W_0(\cdot)$ is the principal branch of the Lambert $W$ function. That is, $y = W_0(x)$ is the solution to the equation $ye^y = x$ with $x \geq 0$. We provide the derivation for Equation (9) in Section B.4.

**Initial plateau**. For any depth $L$, the dynamics in Equation (6) has a stable fixed point at the global minimum, $\alpha = 1$. For deep networks, $L \geq 2$, the network has an unstable fixed point at $\alpha = 0$ in addition to the global minimum. If small initialization, the typical choice for feature learning (Woodworth et al., 2020), is used, the learning dynamics exhibits an initial plateau (Saxe et al., 2014; 2019), corresponding to slow escape from the zero fixed point. The duration of the initial plateau $T$ is approximately

$$T = \tau \ln \frac{1}{\alpha(0)}, \text{ for } L = 2; \qquad T = \frac{\tau}{(1 - \frac{2}{L})\alpha(0)^{1-\frac{2}{L}}}, \text{ for } L \geq 3. \tag{10}$$

The plateau duration $T$ increases when the depth $L$ increases and when the initialization $\alpha(0)$ decreases, as shown in Figure 4. However, the plateau duration in the infinite-depth scalar linear network remains finite, with an upper bound of $T < \tau/\alpha(0)$.

**Convergence rate**. At the end of learning, the linear scalar networks with $L \geq 2$ all converge to the global minimum at a linear rate, according to the dynamics in Equation (6). That is, the total weight $\alpha$ is $\epsilon$-close to the global minimum, i.e. $|1 - \alpha| < \epsilon$, after a time of order $\tau \ln \frac{1}{\epsilon}$.

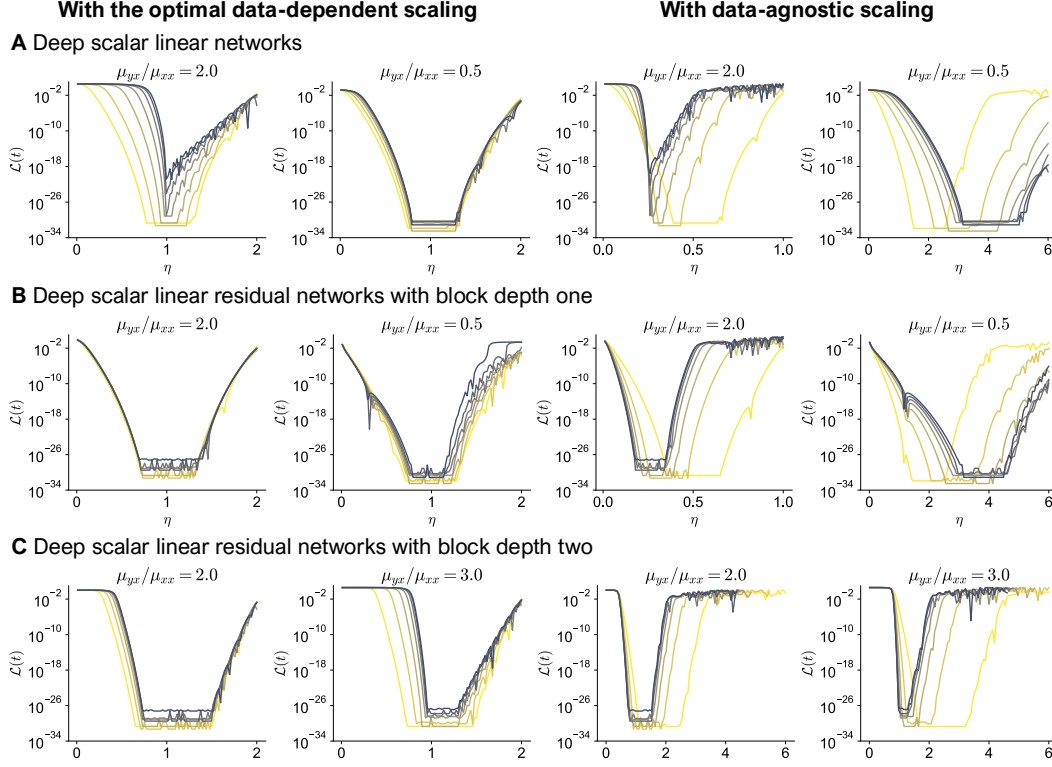

Figure 2: Learning rates transfer under the optimal data-dependent scaling (left two columns), but not under data-agnostic scaling (right two columns). The optimal scaling for deep scalar linear networks, linear residual networks with block depth one and two are given by Equations (5), (28) and (34); the relevant data-agnostic scaling is $\eta \propto L^{-1}, 1, L$, respectively. The loss values are the training loss after 30 steps of gradient descent. The initial weight is set to $w_l(0)^L = 0.1$ in linear networks, and $w_l(0) = 0.01$ in linear residual networks.

## 3 DEPTH-WISE LEARNING RATE TRANSFER

We now examine the implications of the data-dependent learning rate scaling on depth-wise hyperparameter transfer. We show the training loss after a fixed number of gradient descent steps in Figure 2A when the learning rate is scaled as the optimal data-dependent rule in Equation (5) versus as a data-agnostic rule $\eta \propto L^{-1}$. As shown in Figure 2A, the learning rates transfer from a shallower network to deep networks under the optimal scaling, but do not transfer under the data-agnostic scaling. Specifically, the learning rate with $L^{-1}$ scaling for a very deep network is too small when $\mu_{yx}/\mu_{xx} < 1$, and too large when $\mu_{yx}/\mu_{xx} > 1$.

We further extend the analysis to deep scalar linear residual networks with block depth one (Bordelon et al., 2024b; Yang et al., 2024; Marion et al., 2025) and block depth two (Bordelon et al., 2024a; Dey et al., 2025), defined as

$$f_{\text{block1}}(x; w) = \prod_{l=1}^{L} \left( 1 + \frac{w_l}{\sqrt{L}} \right) x, \qquad f_{\text{block2}}(x; w) = \prod_{l=1}^{L} \left( 1 + \frac{w_l^2}{L} \right) x. \qquad (11)$$

Their optimal learning rate scaling rules are calculated in Equations (28) and (34), which are also data-dependent. Similar to deep scalar linear networks, the learning rates transfer under the optimal data-dependent scaling, but does not under data-agnostic scaling, as shown in Figure 2B,C.

On the flip side, we note that the data dependence of optimal learning rate scaling is weak for large $L$. Hence, transferring the optimal learning rate from an intermediate depth to large depth under $L^{-1}$ scaling may still suffice despite being suboptimal, whereas transferring the learning rate from a small depth (e.g. $L = 2, 4$) to large depth would likely fail, as we can see from Figure 2.

In summary, we study depth-wise learning rate scaling in deep scalar linear networks, with and without residual connections, and find that the optimal scaling for transfer is data-dependent.

ACKNOWLEDGMENTS

We thank Kevin Han Huang for feedback on a draft of this paper. We thank the following funding sources: Gatsby Charitable Foundation (GAT4058) to YZ, PEL, LCV, and AS; Sainsbury Wellcome Centre Core Grant from Wellcome (219627/Z/19/Z) to AS; Schmidt Science Polymath Award to AS. AS is a CIFAR Azrieli Global Scholar in the Learning in Machines & Brains program.

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

## A    ADDITIONAL RELATED WORK

A diverse body of theoretical research has investigated neural networks in the limit of large depth, regarding their expressivity (Poole et al., 2016; Hanin, 2019), initialization scheme (Saxe et al., 2014; Schoenholz et al., 2017; Xiao et al., 2018; Yang et al., 2024), the network output at initialization (Hayou, 2023; Noci et al., 2023), the minimum-norm solution (Jacot, 2023; Boix-Adsera, 2025), and formulations based on implicit layers (Amos & Kolter, 2017; Bai et al., 2019) and continuous-depth limits (Haber & Ruthotto, 2017; Chen et al., 2018). Despite this progress, characterizing the behaviors of such deep networks once gradient descent training begins poses a greater challenge. Exact solutions for the full training dynamics have been derived for deep linear networks with aligned small initial weights and whitened data (Saxe et al., 2014; 2019). For nonlinear networks, current findings characterize the gradient descent dynamics over only one or several steps (Jelassi et al., 2023; Hayou, 2023; Bordelon & Pehlevan, 2025; Bordelon et al., 2024a; Chizat, 2025), while the full learning dynamics is generally intractable.

## B    DEEP SCALAR LINEAR NETWORKS

### B.1    ADDITIONAL FIGURES

In Figure 3, we show the trajectories of the total weight with different depths and learning rates in deep scalar linear networks. In Figure 4, we show the trajectories of the total weight and loss with different depths and initialization in deep scalar linear networks.

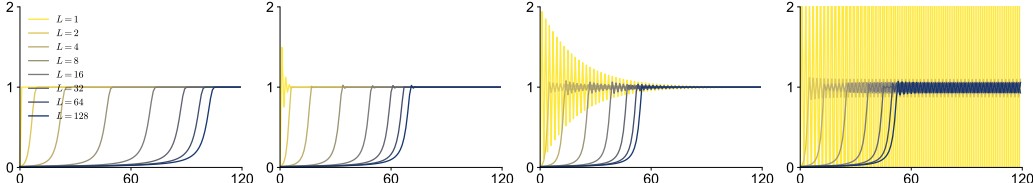

Figure 3: Dynamics of $\alpha(t)$ with different depths $L$ and learning rates $\eta$. The learning rate $\eta$ is given by Equation (5) with $\tau^{-1} = 1, 1.5, 1.95, 2.05$ for the four panels from left to right. When $0 < \tau^{-1} \leq 1$, the gradient descent dynamics is monotonic and well described by the gradient flow dynamics. When $1 < \tau^{-1} < 2$, the gradient descent dynamics is oscillatory but converging. When $\tau^{-1} \geq 2$, the gradient descent dynamics is oscillatory and diverging. Here the initialization is $\alpha(0) = 0.01$. The data statistics are $\mu_{yx} = 1, \mu_{xx} = 1$.

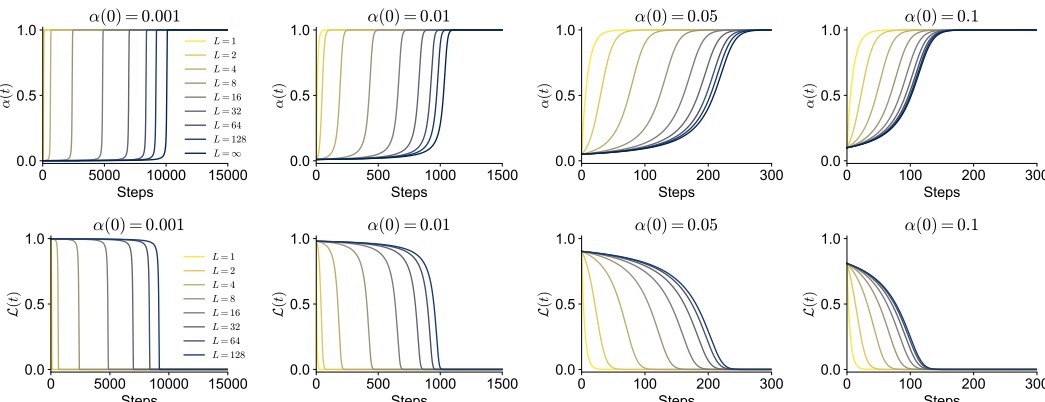

Figure 4: Dynamics of total weights (top row) and loss (bottom row) with different depths $L$ and initialization $\alpha(0)$. The learning speed decreases when the depth increases and when the initialization scale decreases. Here the learning rate $\eta$ is given by Equation (5) with $\tau^{-1} = 0.1$. The data statistics are $\mu_{yx} = 1, \mu_{xx} = 1$.

## B.2 Derivation of the sharpness in Equation (5)

By differentiating the gradient in Equation (3) with respect to $w_1$, we obtain the sharpness of the loss landscape at which the weights in all layers are equal to $w_1$

$$\frac{1}{\eta}\frac{\partial \dot{w}_1}{\partial w_1} = -\mu_{yx}(L-1)w_1^{L-2} + \mu_{xx}(2L-1)w_1^{2L-2}. \tag{12}$$

The sharpness at the global minimum, denoted as $S$, is

$$S \equiv \frac{1}{\eta}\frac{\partial \dot{w}_1}{\partial w_1}\bigg|_{w_1=\left(\frac{\mu_{yx}}{\mu_{xx}}\right)^{1/L}} = \mu_{xx}L\left(\frac{\mu_{yx}}{\mu_{xx}}\right)^{2-2/L}. \tag{13}$$

For $L = 1$, the sharpness depends only on the input variance, $\mu_{xx}$, but not the input-output correlation, $\mu_{yx}$. For $L \geq 2$, the sharpness depends on both the input variance and the input-output correlation. We note that Equation (13) with $\mu_{xx} = 1$ appeared in Saxe et al. (2014, Equation (41)).

Remark: When deriving the gradient descent dynamics, we calculate the negative gradient using the original loss expression with $L$ variables before substituting in the reduction $w_1 = w_2 = \cdots = w_L$. Substituting in the equality before taking the gradient would yield the wrong gradient descent dynamics. However, when calculating the second-order derivative, we differentiate the expression in Equation (3), which is the gradient after substituting in the reduction $w_1 = w_2 = \cdots = w_L$. Substituting in the equality after the double differentiation would yield the wrong sharpness metric. This is because we want the sharpness of the loss landscape along the $w_1 = w_2 = \cdots = w_L$ path, not the sharpness along the $w_1$ axis with the rest of the weights held fixed.

## B.3 Derivation of of the total weight dynamics in Equation (6)

Using Equation (3), we obtain the dynamics of the total weight $a = w_1^L$

$$\dot{a} = \eta L a^{2-2/L}\left(\mu_{yx} - \mu_{xx}a\right). \tag{14}$$

Equation (14) with $\mu_{xx} = 1$ appeared in Saxe et al. (2014, Equation (15)). Substituting the learning rate in Equation (5) into the dynamics of $a$, we get

$$\tau\dot{a} = \left(\frac{\mu_{yx}}{\mu_{xx}}\right)^{-2+2/L} a^{2-2/L}\left(\frac{\mu_{yx}}{\mu_{xx}} - a\right). \tag{15}$$

We denote the total weight divided by the target weight as $\alpha(t) = w_1(t)^L\mu_{xx}/\mu_{yx}$, which represents the relative portion of the target weight learned, with $\alpha = 1$ being the global minimum. The dynamics of $\alpha(t)$ is given by

$$\begin{aligned}
\tau\dot{\alpha} &= \tau\frac{\mu_{xx}}{\mu_{yx}}\dot{a} \\
&= \left(\frac{\mu_{xx}}{\mu_{yx}}a\right)^{2-2/L}\left(1 - \frac{\mu_{xx}}{\mu_{yx}}a\right) \\
&= \alpha^{2-2/L}(1-\alpha).
\end{aligned} \tag{16}$$

We arrive at Equation (6) in the main text.

## B.4 Derivation of the infinite-depth solution Equation (9)

We here solve the learning dynamics with $L \to \infty$, which is given by

$$\tau\dot{\alpha} = \alpha^2\left(1 - \alpha\right). \tag{17}$$

By separating variables and integrating both sides, we obtain

$$\int_0^t \frac{1}{\tau}dt' = \int_{\alpha_0}^{\alpha(t)} \frac{d\alpha'}{\alpha'^2(1-\alpha')} \tag{18}$$

$$\Rightarrow \quad \frac{t}{\tau} = \left(-\frac{1}{\alpha} - \ln\left(\frac{1}{\alpha} - 1\right)\right)\bigg|_{\alpha_0}^{\alpha(t)}. \tag{19}$$

Equation (19) appeared in Saxe et al. (2014, Equation (17)). We rearrange Equation (19) and obtain

$$\frac{1}{\alpha(t)} - 1 + \ln\left(\frac{1}{\alpha(t)} - 1\right) = -\frac{t}{\tau} + \frac{1}{\alpha_0} + \ln\left(\frac{1}{\alpha_0} - 1\right) - 1 \overset{\text{def}}{=} \beta(t). \tag{20}$$

Taking the exponential of both sides yields

$$\left(\frac{1}{\alpha(t)} - 1\right) e^{\frac{1}{\alpha(t)} - 1} = e^{\beta(t)}. \tag{21}$$

Because the principal branch of the Lambert $W$ function, denoted $y = W_0(x)$, solves the equation $ye^y = x$ with $x \geq 0$, we have

$$\frac{1}{\alpha(t)} - 1 = W_0(e^{\beta(t)})$$

$$\Rightarrow \quad \alpha(t) = \frac{1}{1 + W_0(e^{\beta(t)})}, \quad 0 < \alpha \leq 1. \tag{22}$$

We arrive at Equation (9) in the main text.

## C    DEEP SCALAR LINEAR RESIDUAL NETWORKS WITH BLOCK DEPTH ONE

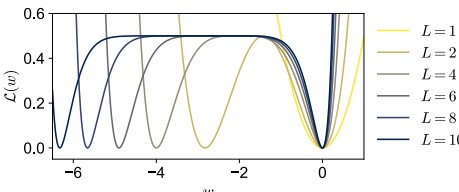

Figure 5: The loss landscape of scalar linear residual networks with block depth one. Similar to the scalar linear chain in Figure 1, the sharpness of the global minimum increases with the depth $L$. Specifically, the plotted curves are $\mathcal{L}(w) = \left(1 - (1 + w/\sqrt{L})^L\right)^2 / 2$, with different $L$.

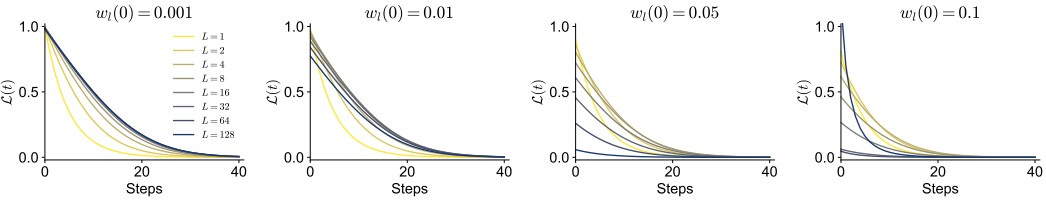

Figure 6: Loss trajectories of deep scalar linear residual networks with block depth one with different depths and initialization. Here the learning rate $\eta$ is given by Equation (28) with $\tau^{-1} = 0.1$. The data statistics are $\mu_{yx} = 2, \mu_{xx} = 1$.

Consider a scalar linear residual network with block depth one defined as

$$f(x; w) = \prod_{l=1}^{L} \left(1 + \frac{w_l}{\sqrt{L}}\right) x, \quad x, w_1, \cdots, w_L \in \mathbb{R}. \tag{23}$$

The $1/\sqrt{L}$ factor is a standard choice consistent with Bordelon et al. (2024b); Yang et al. (2024). The gradient flow dynamics trained with $\ell_2$ loss is given by

$$\dot{w}_1 = \frac{\eta}{\sqrt{L}} \left[\mu_{yx} - \mu_{xx} \prod_{i=1}^{L} \left(1 + \frac{w_i}{\sqrt{L}}\right)\right] \prod_{i \neq l} \left(1 + \frac{w_i}{\sqrt{L}}\right). \tag{24}$$

Similar to the deep scalar linear network, we make the assumption of having equal initial weight in each layer, $w_l(0) = w_1(0) \,\forall l$, which will remain equal throughout training due to the conservation

law. With equal weight in each layer, the gradient flow dynamics reduces to an one-dimensional ordinary differential equation

$$\dot{w}_1 = \frac{\eta}{\sqrt{L}} \left[ \mu_{yx} - \mu_{xx} \left( 1 + \frac{w_1}{\sqrt{L}} \right)^L \right] \left( 1 + \frac{w_1}{\sqrt{L}} \right)^{L-1}. \tag{25}$$

By differentiating the gradient in Equation (25) with respect to $w_1$, we obtain the sharpness of the loss landscape at which the weights in all layers are equal to $w_1$

$$\frac{1}{\eta} \frac{\partial \dot{w}_1}{\partial w_1} = \frac{1}{L} \left[ -\mu_{yx}(L-1) \left( 1 + \frac{w_1}{\sqrt{L}} \right)^{L-2} + \mu_{xx}(2L-1) \left( 1 + \frac{w_1}{\sqrt{L}} \right)^{2L-2} \right]. \tag{26}$$

The sharpness at the global minimum is

$$S \equiv \frac{1}{\eta} \frac{\partial \dot{w}_1}{\partial w_1} \Bigg|_{1+\frac{w_1}{\sqrt{L}}=\left(\frac{\mu_{yx}}{\mu_{xx}}\right)^{1/L}} = \mu_{xx} \left( \frac{\mu_{yx}}{\mu_{xx}} \right)^{2-2/L}. \tag{27}$$

Hence, the maximum stable learning rate scales as

$$\eta = \tau^{-1} \frac{1}{\mu_{xx}} \left( \frac{\mu_{yx}}{\mu_{xx}} \right)^{-2+2/L}, \tag{28}$$

where $\tau \in (0.5, \infty)$ is the time constant.

As shown in Figure 2B, the optimal learning rate transfers under the data-dependent scaling in Equation (28), but does not transfer under the data-agnostic constant scaling of $\eta \propto 1$. Similar to deep scalar linear networks without residual connections, we note that the data dependence of the maximum stable learning rate is weak for large $L$ in deep scalar linear residual networks with block depth one, $\lim_{L\to\infty} 2/L = 0$. Thus, transferring the optimal learning rate from an intermediate depth to infinite depth under the constant scaling is still justified, whereas transferring the learning rate from a small depth (e.g. $L = 2, 4$) to infinite depth would likely fail, as we can see from Figure 2B.

## D DEEP SCALAR LINEAR RESIDUAL NETWORKS WITH BLOCK DEPTH TWO

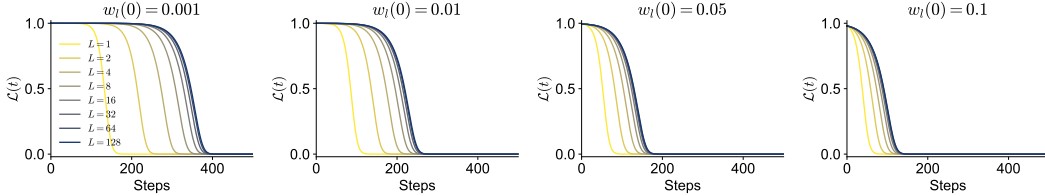

Figure 7: Loss trajectories of deep scalar linear residual networks with block depth two with different depths and initialization. Here the learning rate $\eta$ is given by Equation (34) with $\tau^{-1} = 0.1$. The data statistics are $\mu_{yx} = 2, \mu_{xx} = 1$.

Consider a scalar linear residual network with block depth two defined as

$$f(x; w) = \prod_{l=1}^{L} \left( 1 + \frac{w_l^2}{L} \right) x, \quad x, w_1, \cdots, w_L \in \mathbb{R}. \tag{29}$$

The $1/L$ factor is a standard choice consistent with Bordelon et al. (2024a); Dey et al. (2025). The gradient flow dynamics trained with $\ell_2$ loss is given by

$$\dot{w}_1 = \frac{2\eta w_1}{L} \left[ \mu_{yx} - \mu_{xx} \prod_{i=1}^{L} \left( 1 + \frac{w_i^2}{L} \right) \right] \prod_{i\neq l} \left( 1 + \frac{w_i^2}{L} \right). \tag{30}$$

Similar to the deep scalar linear network, we make the assumption of having equal initial weight in each layer, $w_l(0) = w_1(0) \,\forall l$, which will remain equal throughout training due to the conservation

law. With equal weight in each layer, the gradient flow dynamics reduces to an one-dimensional ordinary differential equation

$$\dot{w}_1 = \frac{2\eta}{L}\left[\mu_{yx} - \mu_{xx}\left(1 + \frac{w_1^2}{L}\right)^L\right]\left(1 + \frac{w_1^2}{L}\right)^{L-1} w_1. \tag{31}$$

By differentiating the gradient in Equation (25) with respect to $w_1$, we obtain the sharpness of the loss landscape at which the weights in all layers are equal to $w_1$

$$\frac{1}{\eta}\frac{\partial \dot{w}_1}{\partial w_1} = \frac{2}{L}\left[-\mu_{yx}\left(1 + \frac{w_1^2}{L}\right)^{L-2}\left(\frac{2L-1}{L}w^2 + 1\right) + \mu_{xx}\left(1 + \frac{w_1^2}{L}\right)^{2L-2}\left(\frac{4L-1}{L}w^2 + 1\right)\right]. \tag{32}$$

The sharpness at the global minimum is

$$S \equiv \frac{1}{\eta}\frac{\partial \dot{w}_1}{\partial w_1}\bigg|_{1 + \frac{w_1^2}{L} = \left(\frac{\mu_{yx}}{\mu_{xx}}\right)^{1/L}} = \frac{4}{L}\mu_{xx}\left(\frac{\mu_{yx}}{\mu_{xx}}\right)^{2-2/L} w^2$$

$$= 4\mu_{xx}\left(\frac{\mu_{yx}}{\mu_{xx}}\right)^{2-2/L}\left(\left(\frac{\mu_{yx}}{\mu_{xx}}\right)^{1/L} - 1\right) \tag{33}$$

Hence, the maximum stable learning rate scales as

$$\eta = \tau^{-1}\frac{1}{4\mu_{xx}}\left(\frac{\mu_{yx}}{\mu_{xx}}\right)^{-2+2/L}\left(\left(\frac{\mu_{yx}}{\mu_{xx}}\right)^{1/L} - 1\right)^{-1} \tag{34}$$

where $\tau \in (0.5, \infty)$ is the time constant.

The scaling of Equation (33) with respect to $L$ is not immediately apparent. To see its behavior with large $L$, we Taylor expand Equation (33) around $1/L = 0$, which yields

$$S = \frac{4}{L}\mu_{xx}\left(\frac{\mu_{yx}}{\mu_{xx}}\right)^2 \ln\left(\frac{\mu_{yx}}{\mu_{xx}}\right) + O\left(\frac{1}{L^2}\right). \tag{35}$$

This shows that the sharpness at the global minimum decreases with $L$, scaling as $1/L$. Therefore, if we were to use a data-agnostic power-law scaling, the learning rate would scale with depth as $\eta \propto L$.

In Figure 2C, we compare the learning rate transfer between the exact maximum stable learning rate scaling in Equation (34) and the data-agnostic scaling of $\eta \propto L$. Similar to the cases with deep scalar linear networks and scalar linear residual networks with block depth one, the optimal learning rate transfers under the data-dependent scaling in Equation (34), but not under the data-agnostic scaling of $\eta \propto L$.

