# OpenReview forum: "Optimal learning rate scaling depends on data in deep scalar linear networks"
_ICLR.cc/2026/Workshop/Sci4DL — Sci4DL 2026_

### Official Review · Reviewer_xvV3 · 2026-02-18

**Fit:** 3
**Significance:** 2
**Confidence:** 2

**Summary:**

This paper studies the optimal learning rate for deep linear scalar networks according to its exact solution, and demonstrates that the optimal learning rate is data-dependent and leads to hyperparameter transfer across depths.

**Strengths:**

The paper provides exact solutions of gradient descent dynamics for deep linear scalar networks and also gives valuable insights concerning the learning rate schedules.

**Suggestions:**

The authors should clarify how their solutions are related to existing solutions on deep linear (diagonal) networks. Moreover, the learning rate schedule is obtained by the stability condition at the global minimum, and it is unclear how this schedule is related to lazy/rich regimes and why it helps hyperparameter transfer.

---

### Official Review · Reviewer_9zMM · 2026-02-22

**Fit:** 3
**Significance:** 3
**Confidence:** 2

**Summary:**

The authors extend the analytic solution of deep scalar linear networks (with and without residual connections), using the extension to show how data dependence is essential for appropriate scaling of the learning rate with network depth.

**Strengths:**

The submission is aimed at an important topic in modern network design, namely scaling behavior with changes in data and network size. The analysis is rigorous, and the presentation is accessible to non-experts as well as those who are active in this particular area.

**Suggestions:**

My most significant concern is the realism of the various assumptions required to derive the main results of the paper. Some assumptions are clearly necessary to make analytic progress, but the extreme symmetry of the current model may lead to regularities that are at odds with real networks. Any argument to the contrary would be useful, for example empirical evidence showing that these assumptions correctly model some aspect of the real world.

---

### Official Review · Reviewer_8Qgt · 2026-02-24

**Fit:** 2
**Significance:** 2
**Confidence:** 2

**Summary:**

This work studies how the optimal learning rate (in the sense of stable convergence) evolves as a function of depth in deep linear networks. They show that, even in this simple case, the optimal learning rate is data-dependent -- specifically, as a function of the second moments of the dataset. Data-agnostic rules for learning rate transfer do not work well across depth, further supporting the data-dependence. However, the learning dynamics are not a function of the data, in both the finite and infinite depth limits.

**Strengths:**

- The research questions and setting are specific and clearly answered.
- The main results are extended to linear networks with residual connections, where the dynamics remain data-agnostic.
- The results are clearly interpreted and there is some discussion on their significance.

**Suggestions:**

- The experimental/simulation setup for the figures throughout the paper is unclear -- some context (even if included in the Appendix) would be very helpful in understanding how well the theory matches experiments.
- It would also be interesting to empirically see if the trends predicted by theory continue to hold with the introduction of non-linear activation functions.
- A brief discussion on how model width (or the combination of width and depth) influences these results would provide a more holistic picture or training dynamics.

Questions:
- The assumption of using equal weights in each layer seems extremely restrictive -- is this an assumption that is commonly used, and is there any way around it? This is just a curiosity -- I expect that things would still be very data-dependent.

---

### Meta-Review · Area_Chair_ZBGF · 2026-03-01

**Recommendation:** Accept

**Metareview:**

This work provides ab analytic solution of deep scalar linear networks (with and without residual connections) and shows how data dependence is essential for scaling of the learning rate with network depth. It is good fit for the workshop and I recommend an accept.

---

### Decision · Program_Chairs · 2026-03-02

Accept